# Transmission Electron Microscopy Observation of Morphological Changes to Cryptophlebia Leucotreta Granulovirus following Ultraviolet Irradiation

**DOI:** 10.3390/pathogens12040590

**Published:** 2023-04-13

**Authors:** Patrick Mwanza, Gill Dealtry, Michael Lee, Sean Moore

**Affiliations:** 1Department of Human Physiology, Nelson Mandela University, Gqeberha 6001, South Africa; 2Centre for HRTEM, Nelson Mandela University, Gqeberha 6001, South Africa; michael.lee@mandela.ac.za; 3Citrus Research International, Gqeberha 6070, South Africa; 4Centre for Biological Control, Department of Zoology and Entomology, Rhodes University, Makhanda 6139, South Africa

**Keywords:** baculovirus, biopesticide, CrleGV-SA, UV susceptibility, occlusion body

## Abstract

Cryptophlebia leucotreta granulovirus (CrleGV), a double-stranded DNA virus (genus *Betabaculovirus*, family *Baculoviridae*), is highly infective to the citrus insect pest *Thaumatotibia leucotreta*. The South African isolate CrleGV-SA is formulated into a commercial biopesticide and registered for use in several countries. In South Africa, it is used as a biopesticide in a multi-faceted integrated pest management approach for citrus crops involving chemical and biological control methods. The virus nucleocapsid is surrounded and protected by an occlusion body (OB) composed of granulin protein in a crystalline matrix. Like all other baculoviruses, CrleGV is susceptible to ultraviolet (UV) radiation from sunlight. This reduces its efficacy as a biopesticide in the field and necessitates frequent respraying. UV damage to baculovirus biopesticides is detected by means of functional bioassays. However, bioassays do not give an indication of whether any structural damage has occurred that may contribute to functional loss. In this study, transmission electron microscopy (TEM) was used to observe damage to the OB and nucleocapsid (NC) of CrleGV-SA, following controlled UV irradiation in the laboratory to mimic field conditions. The resultant images were compared with images of non-irradiated CrleGV-SA virus. TEM images of irradiated CrleGV-SA samples revealed changes to the OB crystalline faceting, a reduction in the size of the OBs, and damage to the NC following UV exposure for 72 h.

## 1. Introduction

Baculoviruses are pathogenic to insects in the orders Lepidoptera, Diptera, and Hymenoptera with a narrow host range and thus do not harm non-target organisms, including humans who consume plant crops [1,2,3]. As a result of this natural relationship with specific insects, baculoviruses provide an environmentally friendly method to combat crop and forest pests [4]. The use of such biopesticides is gaining popularity as chemical pesticides have come under increasingly stringent regulatory pressure imposed by governments to protect the environment [5]; therefore, a number of baculoviruses have been formulated into commercial biopesticides [6].

One such baculovirus is Cryptophlebia leucotreta granulovirus (CrleGV), which is pathogenic to the citrus pest *Thaumatotibia leucotreta* (Meyrick) (Tortricidae: Lepidoptera), commonly referred to as the false codling moth (FCM) [7]. CrleGV has a crystalline proteinaceous occlusion body (OB) that encloses a single virion. This crystalline OB is stable and allows the virus to survive most environmental conditions [8,9]. Two CrleGV isolates, both from South Africa, hence CrleGV-SA, have been registered as biopesticides in South Africa for use on citrus, avocadoes, macadamias, grapes, and other crops by two commercial producers, River Bioscience (Gqeberha, South Africa) and Andermatt Group AG (Grossdietwil, Switzerland) [7,10]. These biopesticides are used as part of the *T. leucotreta* integrated pest management (IPM) programme, a multifaceted approach to controlling *T. leucotreta* [10].

However, baculoviruses are susceptible to the ultraviolet (UV) radiation component of sunlight and hence lose their activity within hours to a few days after sunlight (UV) exposure [11,12,13]. This results in the necessity for farmers to frequently respray the biopesticide to maintain a viable concentration of the virus in the field. In the laboratory, UV damage to the virus is usually indicated by a loss of activity in bioassays with insect larvae, which can take 7–14 days to complete [14,15,16]. No other methods have been reported as being able to directly detect UV damage in baculoviruses.

In this study, transmission electron microscopy (TEM) was used to determine UV damage to the occlusion body (OB) of the South African isolate CrleGV-SA, which may impact the susceptibility of the virus to UV exposure and loss of insecticidal activity. TEM has previously been applied extensively in the study of baculovirus structure [17,18,19]. The capsule-like structure of the baculovirus OB was first reported by Bergold [17]. Subsequently, TEM provided evidence that the OB structure was crystalline in nature [20]. More recently, electron microscopy has been used to characterise the morphology and size of newly discovered baculoviruses [21,22,23]^.^ Additionally, TEM has been used to identify and differentiate between single and multiple nucleopolyhedroviruses on the basis of the number of virions enclosed within the OB [21,24]. Matilainen et al. [25] employed electron microscopy to investigate the mechanism of baculovirus entry into hepatoma cells with Autographa californica multiple nucleopolyhedrovirus as a case study. Scanning electron microscopy (SEM) is used for baculovirus enumeration [19,26,27]. Dhladhla et al. [19] demonstrated that the method was comparable to the enumeration of CrleGV-SA with dark-field light microscopy. However, electron microscopy has rarely been used to determine the effect of UV irradiation on baculoviruses. Brassel and Benz postulated that the selection of a UV-tolerant virus would result in larger OBs or denser nucleocapsids, but electron micrography observation of the morphology and density of OB protein of a selected UV-tolerant strain of the Cydia pomonella granulovirus (CpGV) found no differences in the morphology of the UV-tolerant strain compared with that in the original strain [28]. Thus, it is not clear from the literature whether UV irradiation damages the OB and potentially results in structural alterations in UV-tolerant strains.

Here, we report the effect of UV irradiation on the morphology of the CrleGV-SA OB analysed by TEM. We demonstrated changes in the lattice structure of the OB and the nucleocapsid (NC). These findings indicate that UV irradiation not only damages viral DNA but also has an impact on the OB and NC ultrastructure.

## 2. Materials and Methods

### 2.1. Virus Purification

A standard protocol, according to Jehle et al. [29] with modifications by Moore [30], was used to prepare purified CrleGV-SA. *Thaumatotibia leucotreta* larval cadavers infected with a known isolate of CrleGV-SA were homogenised using a mortar and pestle, filtered through Miracloth, pelleted at 13,000× g for 10 min in a Sigma 3K30 benchtop centrifuge (Sigma Laborzentrifugen GmbH, Osterode am Harz, Germany), and resuspended in 6 mL of 0.1% (*w*/*v*) sodium dodecyl sulphate (SDS, Merck, Johannesburg, South Africa) in double-distilled water. Partially purified viral OBs were centrifuged in an Optima Ultracentrifuge Beckman L70 rotor (Beckman Coulter, Indianapolis, USA) at 40,572× g for 15 min through a continuous gradient of 30–80% (*v*/*v*) glycerol in double-distilled water (Sigma-Aldrich, Johannesburg, South Africa). The resultant virus OB band was removed and washed twice with distilled water by centrifugation for 10 min in the Sigma 3K30 bench top centrifuge at 13,000× g, and the OB pellet was resuspended in 8 mL of sterile distilled water and stored in 500 µL aliquots at 4 °C.

### 2.2. Enumeration of OBs by Dark Field Microscopy

The virus was quantified using a dark field microscopy method described by Hunter-Fujita et al. and modified according to Dhladhla et al. [19,27]. Five microlitres of the virus suspension in 0.1% (*w*/*v*) SDS in double distilled water was counted in a 0.02 mm deep Helber bacterial counting chamber (Hawksley, Lancing, UK) on an Olympus BX 51 TF microscope (Olympus, Tokyo, Japan).

### 2.3. UV Irradiation

UV irradiation was carried out in a Q-Sun Xe-3 HC test chamber (Q-Lab Corporation, Cleveland, USA) fitted with three 100 W xenon arc lamps and a Daylight Q optical filter to provide irradiance at 300 Wm^−2^, which mimicked UV conditions in normal sunlight. A temperature of 30 °C and relative humidity of 42% recreated the field conditions based on the average data collected over one summer period in the Sundays River Valley, Eastern Cape Province, an important citrus growing area (Linta Greeff, Sundays River Citrus Company, personal communication). Purified CrleGV-SA aliquots of 3 mL at a concentration of 1 × 10^10^ OBs/mL were air-dried overnight in Petri dishes under a laminar flow hood and then placed in the UV test chamber for 72 h. The UV-exposed virus samples were resuspended in 3 mL of double-distilled water and quantified and stored at 4 °C until needed.

### 2.4. Preparation of Samples for TEM

The UV-exposed CrleGV-SA and a matched unexposed CrleGV-SA sample were each diluted with double-distilled water to give 1 mL aliquots at a concentration of 1 × 10^8^ OBs/mL and prepared for TEM imaging. A modification of the method described by Wolff et al. [31] was used to prepare the OB samples for TEM. Viral pellets prepared by centrifugation at 3500× g for 5 min were fixed overnight in Karnovsky’s fixative (2.5% glutaraldehyde, 2% paraformaldehyde in 0.05 M pH 7.2 phosphate buffer, and 0.001 M calcium chloride); the fixative was removed by centrifugation at 3500× g for 5 min, and the samples washed and post-fixed in 1% osmium tetroxide in double-distilled water for 1 h at room temperature. Following post-fixation, the excess osmium tetroxide was removed by centrifugation at 3500× g for 5 min, and the samples were washed with double-distilled water and then dehydrated in acetone. The samples were embedded in Spurr’s low-viscosity resin, and the resin block was sectioned using a Leica EM UC7 ultramicrotome (Leica, Wetzlar, Germany). Sections (70 nm thick) were placed on a 300 mesh carbon coated copper grid and stained with drops of 4% uranyl acetate in double-distilled water for 2 min, followed by lead citrate for 1 min. The grids were dried in a vacuum desiccator, and the sections were imaged in a JEOL JEM-2100 TEM (JEOL Ltd, Tokyo, Japan) in bright field (BF) mode. Images were obtained at an accelerating voltage of 200 kV and a low probe current to prevent damage to the specimen.

### 2.5. Data Analysis

For statistical analysis of each sample group (control and UV-exposed CrleGV-SA), 100 OBs were counted in the longitudinal orientation from randomly selected images and classified as intact or damaged. The percentages of unaltered and damaged OBs were recorded and compared.

Measurements of each of the 100 OBs were taken along the length and width of the OB and NC cavity using the image processing software ImageJ^®^ version 2.11.0 (National Institutes of Health, Bethesda, NSW, USA) (Figure 1), and the mean and standard error of each of these measurements were calculated. A paired Student’s t-test (two-tailed) at *p* ≤ 0.05 was conducted to compare the unexposed control and UV-exposed samples. The NC to OB width ratio (NC:OB) was calculated for the OBs in a longitudinal orientation (Figure 1) to determine whether UV irradiation resulted in a change in the size of the area around the NC cavity. The mean and standard error of the ratio were then compared pairwise between the two groups using a Student’s *t*-test.

## 3. Results

### 3.1. TEM Morphology of Unexposed CrleGV-SA

The unexposed control samples observed by TEM displayed the typical crystalline faceting associated with baculovirus OBs and a clearly defined nucleocapsid with a double-layered envelope (Figure 2a,c). The corresponding Fast Fourier Transform (FFT) image confirmed the crystalline nature of the OB (Figure 2b). Direct line profiling intensity measurements gave a lattice line spacing of 6.3 ± 0.1 nm (compared with the FFT measurement of 6.1 ± 0.1 nm). Most of the OBs observed using TEM contained a single NC, although rare occurrences of OBs with double NCs were recorded.

### 3.2. TEM Morphology of UV-Exposed CrleGV-SA

CrleGV-SA samples exposed to UV for 72 h showed signs of damage. The NCs of affected OBs appeared thin, distorted, and in some cases disintegrating (Figure 3a,c). In comparison, the NCs of the unexposed control were of regular shape and normal dimensions (Figure 2). In addition, the double-layered envelope of the NC appeared to be breaking down, and the proteinaceous OB appeared to be disintegrating from the interior outwards in the UV-exposed virus. (Figure 3a,c, brown arrows). The damaged OBs were amorphous and did not show evidence of crystalline faceting when analysed using FFT analysis (Figure 3b,d). Visual evidence supported the suggestion that the UV damage was progressive, as shown in Figure 4a, where OB 1 is at an early stage of degradation and OBs 2 and 3 are at more advanced stages of degradation. Despite the severity of the damage observed in most TEM images, some intact OBs were observed (Figure 4c, green arrow) with crystalline faceting present (Figure 4d).

### 3.3. Measurement of OB Dimensions of UV-Irradiated CrleGV-SA

Random counting of 100 OBs from TEM images obtained from three different TEM samples indicated that 82% of the OBs were damaged after UV exposure for 72 h. The mean length and width (±standard error of the mean (SEM)) (longitudinal section) of the OBs of the unexposed virus were 365.31 ± 4.91 nm and 213.47 ± 3.16 nm respectively, and the mean length and width of the NCs were 210.16 ± 3.89 nm and 48.87 ± 0.97 nm, respectively (Table 1). In comparison, the mean lengths of the OBs of the UV-irradiated virus (301.30 ± 6.03 nm) and the nucleocapsid cavities (182.76 ± 5.56 nm) were significantly shorter than those of the unexposed control virus (*p* = 1.27 × 10^−14^ for OB length and *p* = 1.76 × 10^−5^ for NC length) (Table 1), reflecting 18% undamaged virus with the same dimensions as unexposed virus and 82% damaged virus with reduced dimensions. The mean NC cavity width (58.57 ± 1.11 nm) of the UV-exposed virus was significantly larger than that of the unexposed control virus (*p* = 8.89 × 10^−12^) (Table 1).

The mean NC:OB ratio for the UV-exposed virus samples was 0.33 ± 0.007, which significantly differed from the unexposed control virus ratio of 0.23 ± 0.004 (*p* = 1.015 × 10^−24^) (Figure 5). Thus, UV exposure resulted in the thinning of the OB.

## 4. Discussion and Conclusions

TEM imaging has been used to directly observe OB damage caused by UV irradiation of CrleGV-SA. To our knowledge, this is the first such observation reported for any baculovirus. Dhladhla previously described the use of TEM to determine the crystalline structure of the CrleGV-SA OB [32]. This crystalline structure is formed by the OB protein granulin [33]. Whilst granulin protects the OB from harsh environmental conditions and enables the virus to persist for long periods of time in the soil or similar shaded environments, it has been shown that it cannot provide sufficient protection to the NC against UV irradiation to enable prolonged survival when exposed to sunlight [8,34].

The dimensions measured for the unexposed CrleGV-SA OBs and NCs were consistent with the CrleGV-SA OB measurements made by Dhladhla [32] and consistent with dimensions of other granuloviruses, which have OBs ranging from 300–400 nm in length and 120–300 nm in diameter and virion NCs ranging from 200–300 nm in length and 30–60 nm in diameter [2,21]. The FFT images of the control OBs revealed the expected crystalline structure in the protein matrix. It has previously been shown by X-ray diffraction that NPV polyhedra have a body-centred cubic lattice with unit cells that have 123 symmetry and that CpGV, which is closely related to CrleGV, has 123 symmetry [35,36,37]. X-ray diffraction analysis would be required to confirm the appropriate crystal space group for the granulin CrleGV-SA OBs.

Damage to the CrleGV-SA OBs was observed to take various forms, suggesting a stepwise process. The first type of damage observed was the narrowing of the NC. This could be an early stage of damage before the NC envelope breaks down, leading to the OB losing its structural integrity and disintegrating from the inside outwards. Studies have shown that UV exposure causes DNA damage to baculoviruses [38,39,40,41]. Hence, it is possible that the DNA damage also affects the NC integrity, with resultant disintegration. Another effect of UV irradiation is the disappearance of the NC envelope. At this stage, it is not clear whether the NC envelope collapses before or after NC disintegration. Further evidence that the disintegration of the NC occurs as a multi-step process was given by TEM images showing cross-sections of OBs with varying amounts of NC remaining (Figure 4a). Some OBs were observed without the NC envelope but with a relatively intact virion. Further analysis of these images showed that the crystalline faceting of the granulin OB was absent. The NC:OB ratio obtained for UV-exposed virus agrees with the suggestion that the OB was disintegrating from the centre and progressing outwards. On average, the UV-exposed virus OBs were at least 65 nm shorter than the control unexposed virus OBs.

A distinct characteristic was the loss of the crystalline faceting and lack of lattice fringes of the OBs after UV exposure (Figure 3 and Figure 4). This resulted in the OB becoming amorphous, suggesting that this loss of crystalline faceting could allow UV irradiation to penetrate the nucleocapsid more easily to cause DNA damage to the virion.

This study provided evidence for the effects of UV irradiation equivalent to sunlight exposure on both the CrleGV-SA OB and virion structural integrity. The TEM images of the UV-exposed virus provide a possible indication of the UV damage caused by sunlight in the field situation, which likely leads to the loss of virulence of baculovirus biopesticides. TEM provides a direct rapid technique to differentiate between UV-susceptible and UV-tolerant viruses compared with bioassay. Consequently, it may be possible to determine whether there are differences in the UV sensitivity of different virus isolates of the same species and thus select the isolate that demonstrates the greatest potential for persistence in the field for development as a biopesticide. Several different isolates of CrleGV have been described [42,43,44,45], two of which are commercialised as biopesticides [46,47]. However, one of the other known isolates may prove more UV-tolerant and thus be a better candidate for development as a biopesticide. Furthermore, 18% of OBs showed no signs of UV degradation after 72 h exposure to UV irradiation. Consequently, virus populations may include individuals possessing a higher tolerance to UV irradiation that can be specifically selected for through serial passage after UV exposure [28,47], as was recently reported by us in a laboratory selection procedure from the same isolate of CrleGV-SA described in this paper [48].

## Figures and Tables

**Figure 1 pathogens-12-00590-f001:**
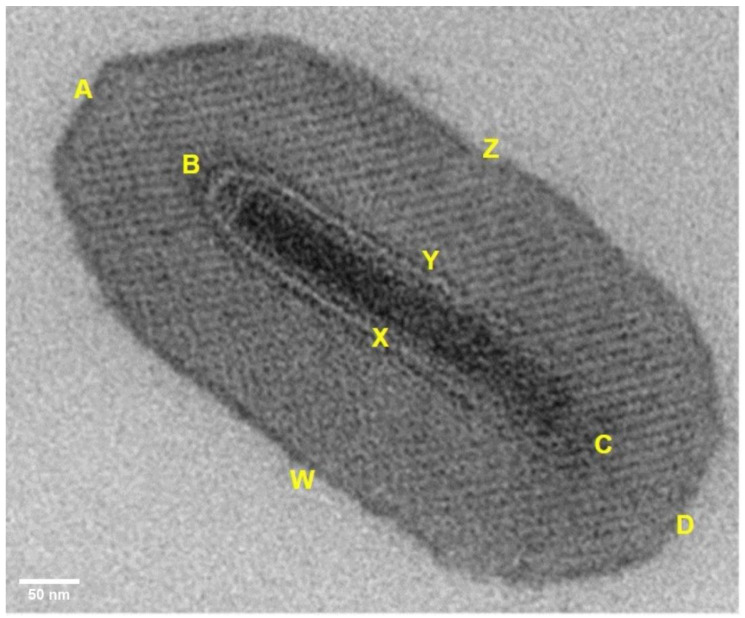
Measurement parameters for the Cryptophlebia leukotreta granulovirus-South Africa (CrleGV-SA) occlusion bodies (OBs) in ImageJ^®^ version 2.11.0. The distance ABCD represents OB length, WXYZ represents OB width, and BC and XY represent nucleocapsid (NC) length and width, respectively. For all OBs measured, the nucleocapsid:occlusion body (NC:OB) ratio was determined using the formula Distance XY ÷ Distance WXYZ. Scale bar = 50 nm.

**Figure 2 pathogens-12-00590-f002:**
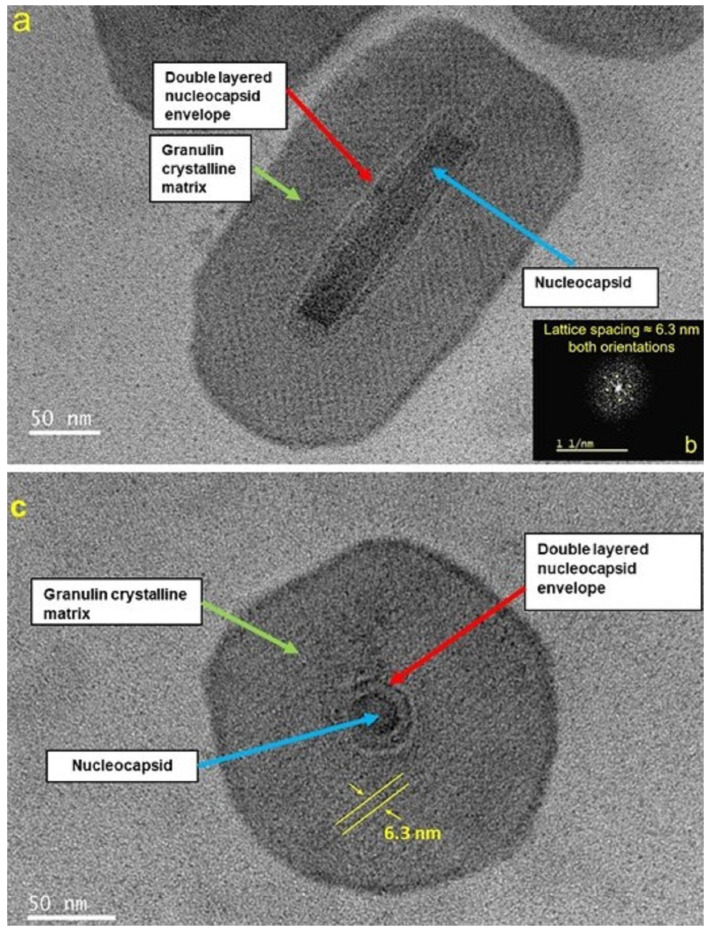
(**a**) Transmission electron microscopy (TEM) longitudinal section of unexposed CrleGV-SA OB with OB granulin crystalline matrix, NC, and double layered envelope of the NC indicated; (**b**) corresponding Fast Fourier Transform (FFT) image for the OB image (**a**) confirming lattice structure and spacing; (**c**) TEM transverse section of unexposed CrleGV-SA OB. Direct measurement of line profiling intensity gave a lattice line spacing of 6.3 ± 0.1 nm (**c**). FFT for (**c**) was identical to that shown in (**b**), not shown.

**Figure 3 pathogens-12-00590-f003:**
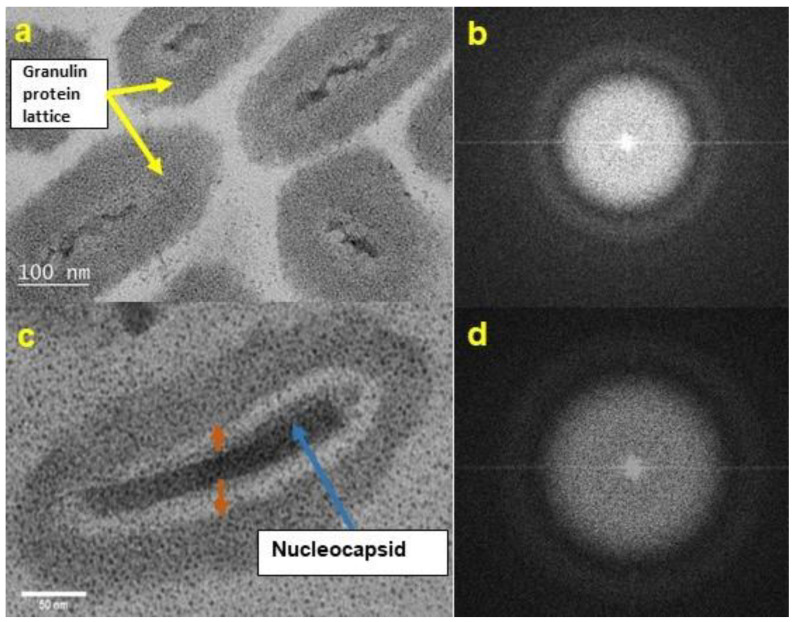
TEM bright field (BF) image of cross-sectional views of ultraviolet (UV)-damaged CrleGV-SA OBs after exposure to UV irradiation for 72 h. (**a**) The crystalline structure of the OB is not visible and the NC appears thinner and disintegrated. NC double envelope is not visible; (**b**) The corresponding FFT image for the OB, top right (**a**), confirming lack of crystalline lattice structure; (**c**) the OB appears to be disintegrating outwards, as indicated by brown arrows; and (**d**) the corresponding FFT image for (**c**) shows no crystalline lattice structure.

**Figure 4 pathogens-12-00590-f004:**
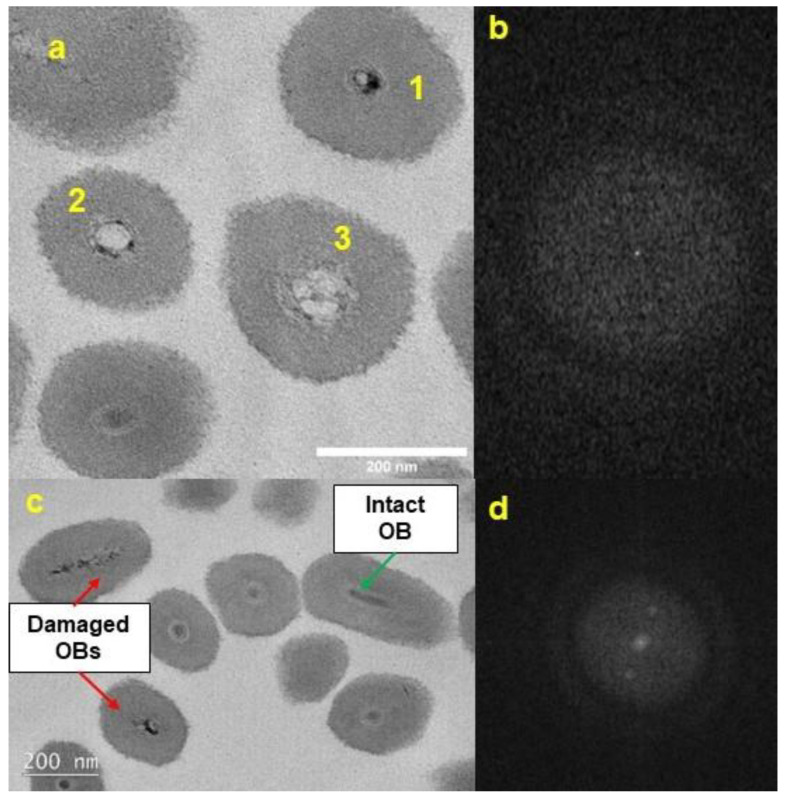
TEM BF images of CrleGV-SA OBs exposed to UV for 72 h, indicating (**a**) progressive degradation of the NC (from 1–3) shown in plan view; (**b**) FFT confirmation for lack of crystalline lattice structure of OBs in (**a**); (**c**) damaged OBs (red arrows) and presence of an intact OB (green arrow) shown in cross-section; (**d**) FFT indication of crystalline lattice structure of intact OB in (**c**).

**Figure 5 pathogens-12-00590-f005:**
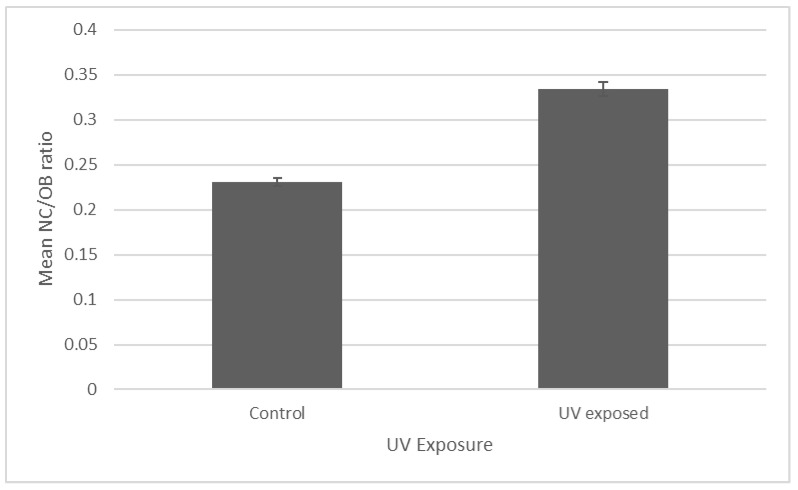
Comparison of the mean NC:OB width ratio. The error bars represent SEM. For each treatment, n = 100 viruses from randomly selected images. The non-UV-exposed control and UV-exposed mean widths differed significantly, *p* ≤ 0.05.

**Table 1 pathogens-12-00590-t001:** Mean (±standard error of the mean, SEM) occlusion body (OB) and nucleocapsid (NC) dimensions of 100 unexposed control Cryptophlebia leukotreta granulovirus-South Africa (CrleGV-SA) and 100 ultraviolet (UV)-exposed CrleGV-SA OBs.

Dimension (nm)	Control Non-Exposed CrleGV-SA ^1^	UV Exposed CrleGV-SA ^1^
OB length	365.31 ± 4.91	301.30 ± 6.03
OB width	213.47 ± 3.16	184.03 ± 3.60
NC length	210.16 ± 3.89	182.76 ± 5.56
NC width	48.87 ± 4.89	58.57 ± 1.11

^1^ All pairwise comparisons between control and exposed virus dimensions differed significantly from one another (*p* < 0.05).

## Data Availability

The data presented in this study are available in Mwanza, P. Development of a UV-tolerant strain of the South African isolate of Cryptophlebia leucotreta granulovirus for use as an enhanced biopesticide for Thaumatotibia leucotreta control on citrus. PhD thesis, 2020, Nelson Mandela University, Gqeberha, South Africa.

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
