# Peer review of "Transmission Electron Microscopy Observation of Morphological Changes to Cryptophlebia Leucotreta Granulovirus following Ultraviolet Irradiation"

_pathogens, 2023, doi:10.3390/pathogens12040590_

Round 1
Reviewer 1 Report
In the present study, transmission electron microscopy (TEM) was used to determine UV damage to the occlusion body of the South African isolate CrleGV-SA, which may impact on susceptibility of the virus to UV exposure and loss of insecticidal activity.
The TEM images of UV exposed virus show impressively the UV damage caused by simulated sunlight in the lab, which likely leads to loss of virulence of baculovirus biopesticides in the field situation as well. TEM provides a direct rapid technique to differentiate between UV-susceptible and UV-tolerant viruses compared to bioassay and should therefore be used as an additional tool. As the authors stated, I agree that consequently, it may be possible to determine whether there are differences in UV-sensitivity of different virus isolates of the same species and thus select the isolate that demonstrates the greatest potential for persistence in the field for development as a biopesticide.
The manuscript is written in a clear and excellent style, no changes are necessary.
One mistake has to be corrected:
Line 178: brown arrows (instead of white arrows)
Author Response
We are grateful for the very positive comments of reviewer 1 who stated that “The TEM images of UV exposed virus show impressively the UV damage caused by simulated sunlight in the lab, which likely leads to loss of virulence of baculovirus biopesticides in the field situation as well” and that “I agree that consequently, it may be possible to determine whether there are differences in UV-sensitivity of different virus isolates of the same species and thus select the isolate that demonstrates the greatest potential for persistence in the field for development as a biopesticide.” The reviewer had just one correction: brown arrows (instead of white arrows) in the legend to figure 3. This has been corrected in line 183.Reviewer 2 Report
The manuscript “Transmission Electron Microscopy Observation of Morphological Changes to Cryptophlebia Leucotreta Granulovirus following Ultraviolet Irradiation” by Mwanza et al. is an important study comparing UV-treated and non-treated Cryptophlebia leucotreta granulovirus (CrleGV). By mimicking the field conditions with controlled UV exposure in a laboratory setup, combined with the TEM imaging technique, this study reveals key structural damage of occlusion body and nucleocapsid upon UV treatment, providing evidence for the loss of function, which is consistent with functional bioassays.
Generally, the manuscript reads soundly and the experimental design is logical. However, revisions are requested as below:
“UV damage to baculovirus biopesticides is detected by means of functional bioassays. However, bioassays do not give an indication of what structural damage has occurred.” This part needs to rewrite, as structural damage is not necessarily directly linked to function loss. Rather, a causal effect relationship needs to be established.
The scale bars are missing on TEM image Figure 1, though this is an example image for measurements. TEM images are challenging to be interpreted without scale bars. The scale bars must be added and properly indicated in the figure legends. When scale bars are present, it
Figure legends need to be revised, especially Figure 3c, where white arrows are not present in Figure 3c.
The quality of the TEM images needs to be enhanced. For example, images in Figures 4a and 4c are of low quality as indicated by their corresponding FFT (Figures 4b and 4d) showing astigmatism, as they are not spherical compared to Figure 3b/d.
Does Figure 2b show a Fast Fourier Transform (FFT) of Figure 2a? Or just a subset of the OB granulin crystalline matrix area? Could the authors indicate the spacing of the crystalline matrix based on the FFT for extra information for the readers? And refer to the existing literature on the spacing of relevant granulin crystalline matrix.
As TEM imaging is a major tool in this paper, more representative images of each condition will be necessary for readers to judge the differences in structural changes. I would recommend a panel of at least 3 images for certain experiments. This can be supplied as sub-panels in the same major figures, or a separate figure just to list the panels, so that the structural differences can be better appreciated.
Author Response
We acknowledge the positive comments of reviewer 2, stating that our manuscript “is an important study comparing UV-treated and non-treated Cryptophlebia leucotreta granulovirus (CrleGV)” and that it “reveals key structural damage of occlusion body and nucleocapsid upon UV treatment, providing evidence for the loss of function, which is consistent with functional bioassays”. We address their specific points below: 1. The statement in the abstract “UV damage to baculovirus biopesticides is detected by means of functional bioassays. However, bioassays do not give an indication of what structural damage has occurred” has been revised to reflect reviewer 2’s comment that “structural damage is not necessarily directly linked to function loss. Rather, a causal effect relationship needs to be established.” The statement in lines 21 and 22 now reads: “However, bioassays do not give an indication of whether any structural damage has occurred that may contribute to functional loss.” 2. “The scale bars are missing on TEM image Figure 1.” A scale bar of 50nm has been added to figure 1 and the figure legend (line 139) has been amended to include “Scale bar = 50 nm” 3. “Figure legends need to be revised, especially Figure 3c, where white arrows are not present in Figure 3c.” Figure 3 figure legend has been corrected and clarified (lines 179-184) to read “Figure 3. TEM BF image of cross sectional views of UV damaged CrleGV-SA OBs after exposure to UV irradiation for 72 h. a) The crystalline structure of the OB is not visible and the NC appears thinner and disintegrated. NC double envelope is not visible; b) The corresponding FFT image for the OB top right (3.a), confirming lack of crystalline lattice structure; c) The OB appears to be disintegrating outwards as indicated by brown arrows; and d) The corresponding FFT image for (3.c) shows no crystalline lattice structure.” 4. “The quality of the TEM images needs to be enhanced. For example, images in Figures 4a and 4c are of low quality as indicated by their corresponding FFT (Figures 4b and 4d) showing astigmatism, as they are not spherical compared to Figure 3b/d” The images for figure 4 have been checked. Images 4a, 4c, and 4d were not astigmatic in the original image. For image 4b the aspect ratio was found to be compromised during processing of the figure and the image 4b has been reinserted with the correct aspect ratio and is not astigmatic. As for the quality of figure 4a and 4c, it is not clear from the reviewer’s comment as to what is meant by poor image quality. However, both images are lower magnification than previous images in this manuscript, and intentionally acquired under conditions of de-focus (under or over) to highlight the damage to the virions. 5. Does Figure 2b show a Fast Fourier Transform (FFT) of Figure 2a? Or just a subset of the OB granulin crystalline matrix area? Could the authors indicate the spacing of the crystalline matrix based on the FFT for extra information for the readers? And refer to the existing literature on the spacing of relevant granulin crystalline matrix. Figure 2b shows an FFT of figure 2a. The figure and legend have been amended to clarify this. Figure 2b showing the FFT has been inserted into the figure 2a, with annotation “Lattice spacing = 6.3 nm both orientations.” Direct measurement of line profiling intensity has been added to figure 2c. The figure legend (line 160-164) now reads “Figure 2. a) TEM longitudinal section of unexposed CrleGV-SA OB, with OB granulin crystalline matrix, NC and double layered envelope of the NC indicated; b) Corresponding FFT image for the OB image (2.a), confirming lattice structure and spacing; c) TEM transverse section of unexposed CrleGV-SA OB. Direct measurement of line profiling intensity gives a lattice line spacing of 6.3 ± 0.1 nm (2.c). FFT for (2.c) was identical to that shown in (2.b), not shown.” The text in lines 154-156 of the results section has been amended to include the sentence “Direct line profiling intensity measurements give a lattice line spacing of 6.3 ± 0.1 nm (compared to FFT measurement of 6.1 ± 0.1 nm).” The existing literature on the spacing of relevant granulin crystalline matrix was already discussed in lines 224-233 in the discussion and conclusion section. 6. As TEM imaging is a major tool in this paper, more representative images of each condition will be necessary for readers to judge the differences in structural changes. I would recommend a panel of at least 3 images for certain experiments. This can be supplied as sub-panels in the same major figures, or a separate figure just to list the panels, so that the structural differences can be better appreciated. We consider that sufficient representative images of UV irradiated virus have been included in 4 separate TEM images from different preparations (figures 3a, 3c, 4a, 4c), showing a total of 16 complete OBs. Figure 4a shows in cross section progressive stages of damage to the OBs and figures 3a, 3c and 4c show longitudinal sections of damaged OBs. Furthermore in table 1 and figure 5 the dimensions of 100 UV exposed and 100 unexposed viruses obtained from TEM images are presented.